# Differences in Meat Quality of Six Muscles Obtained from Southern African Large-Frame Indigenous Veld Goat and Boer Goat Wethers and Bucks

**DOI:** 10.3390/ani12030382

**Published:** 2022-02-04

**Authors:** Gertruida L. van Wyk, Louwrens C. Hoffman, Phillip E. Strydom, Lorinda Frylinck

**Affiliations:** 1Department of Animal Sciences, University of Stellenbosch, Private Bag X1, Matieland, Stellenbosch 7602, South Africa; louisa.vanwyk@virbac.com (G.L.v.W.); louwrens.hoffman@uq.edu.au (L.C.H.); pestrydom@sun.ac.za (P.E.S.); 2Centre for Nutrition and Food Sciences, Queensland Alliance for Agriculture and Food Innovation (QAAFI), The University of Queensland, Coopers Plains, QLD 4108, Australia; 3Agricultural Research Council—Animal Production, Private Bag X2, Irene 0062, South Africa

**Keywords:** Cape Lob Ear, Cape Speckled, Boer Goat, meat goat breeds, meat tenderness, meat colour, collagen, chevon

## Abstract

**Simple Summary:**

This study describes the meat tenderness and colour attributes of six muscles (*Longissimus thoracis et lumborum* (LTL), *Semimembranosus* (SM), *Biceps femoris* (BF), *Supraspinatus* (SS), *Infraspinatus* (IS), *Semitendinosus* (ST)) from same-aged young Boer Goat (BG) and Indigenous Veld Goat (IVG: Cape Speckled and the Cape Lob Ear) wethers and bucks. Muscle tenderness and colour characteristics differed more between wethers and bucks than between IVG and BG. Large-frame IVG bucks and wethers produced very similar meat tenderness, juiciness and colour characteristics to the BG bucks and wethers, indicating them to be just as suited for meat production. The wethers′ meat, with its increased intramuscular fat in all six muscles tested, would satisfy the consumer segment that prefer juicier and more flavoursome meat. Knowledge of the muscle characteristics of goat carcasses will help the development of the formal commercial market for goat meat, which would benefit smallholder farmers, who typically produce most of the goats in the world.

**Abstract:**

Various meat quality characteristics of six muscles (*Longissimus thoracis et lumborum* (LTL), *Semimembranosus* (SM), *Biceps femoris* (BF), *Supraspinatus* (SS), *Infraspinatus* (IS), *Semitendinosus* (ST)) from large-frame Boer Goats (BG) and Indigenous Veld Goats (IVG: Cape Speckled and the Cape Lob Ear) were studied. Weaner male BG (*n* = 18; 10 bucks and 8 wethers) and IVG (*n* = 19; 9 bucks and 10 wethers) were raised on hay and natural grass, and on a commercial pelleted diet to a live weight of 30–35 kg. All goats were slaughtered at a commercial abattoir and the dressed carcasses were chilled at 4 °C within 1 h post mortem. The muscles were dissected from both sides 24 h post mortem and aged for 1 d and 4 d. Variations in meat characteristics such as ultimate pH, water holding capacity (WHC), % purge, myofibril fragment length (MFL), intramuscular fat (IMF), connective tissue characteristics, and Warner-Bratzler shear force (WBSF) were recorded across muscles. Bucks had higher lightness (L*) and hue-angle values, whereas wethers had increased redness (a*) and chroma values. The muscle baseline data will allow informed decisions to support muscle-specific marketing strategies, which may be used to improve consumer acceptability of chevon.

## 1. Introduction

IVG are a group of specific pure-bred indigenous eco-types represented by the IVG-Association, which defines specific standards that a goat must adhere to before it can be classified as one of the eco-types such as the Cape Lob Ear and the Cape Speckled [1]. Both of these eco-types have large frames and can compete with the Boer Goat (BG) in terms of meat yield [2], whilst also having additional advantages such as adaptability to harsh climates and disease resistance [3]. The increasing global human population and the threat of global warming makes it important to promote the production of goat meat (chevon) from adapted eco-types such as the IVG. Although chevon is popular amongst the greater population of southern Africa, chevon is not available on commercial shelves in South Africa, mainly because there are insufficient commercial slaughter numbers to ensure a constant supply to the commercial retail market. Although southern Africa has relatively large numbers of meat goats (703,892 head) [4], most are produced in the informal sector and traded within this sector, thereby making it challenging to obtain official statistics of the volumes of goat meat produced and traded. Available goats are either sold alive for local traditional slaughtering practices or exported to Middle Eastern and Asian countries. Small and emerging southern African farmers are interested in IVGs as they do not require intensive management to be productive. For chevon, quality fresh meat is the most economically profitable; however, scientific knowledge on the meat quality of these breed types is scarce, compared to that of the well-known “improved” BG breed and the undefined “indigenous” goats that are usually used in comparative studies [5,6,7,8,9,10].

The term “meat quality” includes many attributes; of these, texture, juiciness, flavour and visual appeal are important to consumers. Tenderness and the mechanical properties of meat are influenced by the connective tissue, myofibrils and their interactions, which differ between muscles [11,12]. Compared to sheep and cattle, knowledge of the meat quality of BG and large-frame IVG of South Africa is limited due to a previous lack of interest. The goat carcass consists of over a hundred different muscles with different properties, which affect processing characteristics and could influence consumer acceptability [13]. There has been a continued trend in the retail sector to separate muscles, based on perceived connective tissue characteristics, to better market them and apply the knowledge in terms of the users′ requirements. Notable studies on the physical and compositional traits of BG muscles have been conducted over the years [7]. These include carcass measurements and commercial yields [14], as well as cooking and juiciness related quality characteristics [15], including studies to understand the impact of carcass handling on the texture, mainly determined by the Warner-Bratzler shear force (WBSF) on different muscles [8,9,16,17]. Most studies evaluating chevon are conducted on the LTL and SM muscles in terms of tenderness and sensory quality attributes [5,6,9,10]. To establish a baseline for IVG eco-types, this paper focuses on the effect of breed (IVG vs. BG) and castration (Sex: bucks and wethers) on: ultimate muscle pH (pH_u_), % purge, water holding capacity (WHC), WBSF, myofibril fragment length (MFL), intramuscular fat representing marbling, collagen characteristics, and meat colour in six different muscles (i.e., LTL, SM, BF, SS, IS, and ST) to establish baselines for these eco-types.

## 2. Materials and Methods

### 2.1. Animal and Experimental Design

This research was approved by the Agricultural Research Council-Animal Production (ARC-AP) Ethics Committee (ref no. APIEC16/021). Weaner BG (*n* = 18; 10 bucks and 8 wethers) and large-frame IVG (*n* = 19; 9 bucks and 10 wethers) were purchased from several commercial breeders at three months of age (17 kg on average for IVG and 20 kg on average for BG). The sourcing of animals from different producers provided sufficient representation of genetic variation for each breed type. When bought, the commercial breeders had already castrated the male animals on the farm. The animals were reared at the Small Stock Section of the ARC-AP situated in Irene in the Gauteng province of South Africa, where they grazed a natural grass diet supplemented with *Eragrostis curvula* hay (estimated crude protein 48.9 g/kg dry matter (DM); neutral detergent fibre 746 g/kg DM) ad libitum and an average of 250 g commercial “Ram, lamb and ewe-13” pellets (protein 130 g/kg, fat 25–70 g/kg, fibre 150 g/kg, moisture 120 g/kg, calcium 15 g/kg, phosphorus 3 g/kg, urea 10 g/kg; Meadow Feeds, Lanseria, South Africa) per day per animal. The goats were fed for, on average, 6 to 8 months until they attained a live weight (LW) of between 30 and 35 kg. After weighing (LW), the goats were transported for 3 km to the abattoir of the ARC-AP on the day of slaughter. The experimental design is presented in Figure 1 and has been described in more detail in an earlier paper [2]. The carcasses were subjected to electrical stimulation (ES 20 s, 400 Volts peak, 5 ms pulses at 15 pulses/s), 10 min after stunning and exsanguination, after which all the carcasses were placed in the chiller at 4 °C within 60 min of post mortem. After chilling (24 h, <4 °C), the carcasses were removed from the chiller and the specific muscles removed from both sides of the carcass and cut into various slices for the different meat quality analyses (Figure 2). Temperature and pH values were measured 24 h post mortem (pH_u_) on the same chilled muscles used for colour measurement with a calibrated CyberScan PC 300 (Eutech Instruments Pte Ltd., Queenstown, Singapore).

### 2.2. Laboratory Analysis

For the chemical and physical analyses, samples were taken from the various locations of the six muscles, LTL, SM, BF, SS, IS, and ST, as described in Figure 2. Analyses were either conducted on the fresh samples (% purge, WHC, chemical and meat colour analyses) or on vacuum-packed frozen (−20 °C) and then defrosted (4 °C, 24 h) samples such as WBSF.

#### 2.2.1. Purge and Water Holding Capacity

Purge percentage was measured using a 10 mm-thick slice of the six different muscles (LTL, SM, BF, SS, IS, and ST), vacuumed and aged for 4 d at 4 °C. The specific slices were weighed before and after storage and the weight difference indicated as purge loss percentage. The WHCs of the six fresh muscles were determined using the filter paper press method [18]. Briefly, 400 to 500 mg of meat sample was placed on Whatman 4 filter paper, (Camlab Ltd, Cambridge, UK), contained between two Perspex plates. Constant pressure was applied using a hand-operated screw for 5 min. The borders of the meat and fluid were marked out and their areas measured using a video image analyser (Soft Imaging System, Olympus, Tokyo, Japan), according to [19]. WHC was expressed as a ratio of meat area to fluid area.

#### 2.2.2. Warner-Bratzler Shear Force

The frozen vacuum-packed muscle samples (LTL, SM, BF, SS, IS, and ST) were placed in a cold room at 4 °C to thaw for 24 h before cooking. Whole cuts were prepared according to an oven-broiling method using direct radiant heat [20]. Calibrated electric ovens (Mielé ovens, model H217, Miele & Cie. KG, Gütersloh, Germany) were set to “broil” 10 min prior to cooking at 160 °C. The LL samples were placed on an open casserole pan on a rack with no added water (dry cooking). The SM, BF, SS, IS, and ST, were placed in a casserole pan, adding 100 ml water and close with lid (moisture cooking). The cuts were broiled for approximately 20 min until they reached an internal core temperature of 70 °C. The internal temperature was monitored by placing an iron-constant thermocouple (T-type) (Hand-model Kane-Mane thermometer, Kane International Ltd., Hertfordshire, UK) in the approximate geometric centre of each sample. The cooked meat was weighed together with the pan and drip. The cooked samples were cooled for 2 h at room temperature (20 °C) before shear force measurement. Six cylindrical samples (12.5 mm core diameter) were bored parallel to the direction of the muscle fibres. Each core was sheared perpendicular to the myofibrils using a Warner-Bratzler device fitted to an Instron Universal Testing Machine (Model 4301, Instron Ltd., Buckinghamshire, UK) at a crosshead speed of 200 mm/min with one shear in the centre of each core [21]. The toughness of the meat was the average maximum force measured in Newton (N) required to shear through the cores.

#### 2.2.3. Myofibril Fragment Length

Samples used for MFL were aged for 1 d and 4 d post mortem. Sub-samples of approximately 3 g were taken, blended with a blunt blade in cold potassium phosphate extraction buffer at 4 °C to arrest any further proteolysis [22], and determined according to [23]. The droplets of extracted MFL solution were mounted on slides, covered with a cover slip, and viewed under a microscope attached to a video image analysis (VIA). One hundred myofibril fragments per sample were examined and measured at a magnification of 40×.

#### 2.2.4. Chemical Composition and Collagen Characteristics

The protein and IMF (associated with marbling) were analysed using the procedures of the Association of Official Analytical Chemists [24] at the ARC-AP Analytical Laboratories. Samples (25 g of homogenized meat) were freeze-dried according to method 934.01 [24]. The percentage fat content was determined on 5 g of freeze-dried sample using a 1:2 chloroform/methanol solution for fat extraction (SOXTEC method) as described in [25]. The total nitrogen content in the defatted muscle samples was determined after samples had been digested in a micro Kjeldahl system (Analytical Laboratory ARC-AP). The nitrogen content was multiplied by a factor of 6.25 in order to obtain the protein content of the sample, which was subsequently converted to a value per gram of wet meat (method 922.15) [24]. Soluble, insoluble and total collagen were determined in the same fresh samples.

Total collagen content in the six muscles (LTL, SM, BF, SS, IS, and ST) was determined by measuring the total hydroxyl-proline nitrogen content in the hydrolysed samples according to a modified method of [26]. Approximately 1 g of fresh sample was weighed into a hydrolysed tube and mixed with 15 mL of 6 N HCl. The samples were hydrolysed at 120 °C for 16 h, then 0.5 g active carbon was added to each tube, stirred, and filtered through Whatman 4 filter paper. The aliquots were collected in a 100 mL volumetric flask and filled up with distilled water. An aliquot of 50 mL was used for the determination of total collagen, described below.

The solubility of the muscle collagen (hydroxy-proline nitrogen content of soluble collagen) was determined according to the method of [27], with some modifications. About 2 g of fresh sample was stirred in 10 mL of 1% NaCl solution. The samples were heated in a shaking water bath at 78 °C for 60 min. The cooled samples were centrifuged at 10,000 rpm for 15 min. The supernatants were poured into hydrolysing tubes, marked as soluble. The pellet was poured into another hydrolysing tube and marked insoluble. To each tube, 7.5 mL of 6 N HCl (19.2%) was added and hydrolysed overnight at 120 °C. The following day, 0.5 g of active carbon was added to the cooled tubes, stirred, and the homogenates filtered into 50 mL volumetric flasks and filled to the mark with distilled water. Aliquots of 50 mL were used for determination of both soluble and insoluble collagen.

Hydroxy-proline concentrations were determined calorimetrically according to a modified method of [28]. About 1 mL of the final sample was added into the test tubes, to which 1 mL of 10% KOH solution was added (to neutralise the acid in the sample). A blank consisting of 2 mL distilled water was prepared. Standard solutions were prepared containing 0 to 7.5 µg/mL and 2 mL hydroxy-proline to create a new standard curve for each analysis session.

To each test tube (including standards and blanks), 1 mL of the oxidant solution (1.41 g Chloramine-T in a 100 mL, pH 6.8 buffer solution consisting of: 26 g citric acid monohydrate, 14 g sodium hydroxide, 78 g Anhydrous sodium acetate and 250 mL propan-1-ol) was added. The tubes were vortexed for 5 s and left for 20 min at room temperature. After 20 min, 1 mL of the colour reagent (10 g para-dimethylaminobenzaldehyde, 35 mL perchloric acid solution (60%), 65 mL propan-2-ol, prepared fresh) was added and the tubes vortexed. The tubes were heated to 62 ± 5 °C for 30 min, then vortexed. Thereafter, they were cooled to room temperature (a strong, aromatic, pink liquid with a white salt residue formed in the tubes). The top transparent pink liquid was pipetted into disposable micro cuvettes and the absorbance was read on a spectrophotometer at 558 nm (±2 nm). Hydroxy-proline content was determined from the standard addition curve.

Total collagen content was determined by calculating hydroxy-proline nitrogen from hydroxy-proline (molecular mass 131.13 and nitrogen atom number 14.0067). Collagen values were expressed as mg collagen/g of muscle sample by using the hydroxy-proline conversion of 7.25 and 7.53 for insoluble and soluble collagen respectively [29].

#### 2.2.5. Measurement of Colour and pH

The colours of muscle samples (ca. 15 mm thick) were measured fresh at 1 d and 4 d post mortem. The meat samples were allowed to bloom for 60 min at ± 4 °C before the meat colour values were recorded. A Konica-Minolta 600d spectrophotometer (Konica-Minolta Inc. Osaka, Japan) with the software package Spectra Magic NX Pro was used to measure surface D65 at three different positions on the meat samples. Three components were recorded according to the CIELAB colour space (*L*a*b**) defined by the International Commission on Illumination in 1976; lightness, *L** (dark (0) to light (100)) and the two chromatic components; *a** (green (−60, 180°) to red (+60, 0°)) and *b** (blue (−60, 270°) to yellow (+60, 90°)) which represented the myoglobin levels in the meat [30]. The spectrophotometer configuration consisted of illuminate D65 with an observer angle of 10°, and the spectral component excluded (SCE) mode after calibration using a white reference [31]. Chroma (saturation index (S) = (*a**^2^ + *b**^2^)^1/2^ [32] and hue-angle (discolouration) = tan^−1(*b*/a**)^ [33] were calculated from *a** and *b** values; chroma measures colour intensity, where higher values indicate a more intense red colour in meat. An increase in hue-angle between 0° and 90° corresponds to a blending of yellowness or less redness, probably due to metmyoglobin formation in fresh meat. Ultimate pH (pH_u_) was measured with a portable pH meter (Eutech Instruments, Cyber Scan pH 11, Keppel Logistic, Singapore) in the same location in the LTL, SM, BF, SS, IS, and ST muscles at 24 h post mortem.

#### 2.2.6. Statistical Analysis

The data were subjected to analysis of variance [34] to test the effect of breed (BG and IVG), and sex type (bucks and wethers) on six muscles for the following characteristics; pH and temperature (24 h post mortem, pH_u_ and T_u_), WHC (1 and 4 d post mortem), % purge, WBSF (1 and 4 d post mortem), MFL (1 and 4 d post mortem), connective tissue characteristics, and meat colour (*L**, *a**, *b**, chroma and hue-angle, 1 and 4 d post mortem) [35]. Statistical significance (Fisher′s t-test, least significant difference) was calculated at a 5% level to compare means. A value of *p* ≤ 0.05 was considered statistically significant, although in some instances, data with a *p* ≤ 0.1, (10% level) was considered as a trend worth discussing.

Prior to analyses, a Shapiro-Wilk test for normality was performed on the data [36] and, where applicable, outliers (classified as such when the standardized residual for an observation deviated by more than three standard deviations (SD) from the model value) were removed. Where applicable, the closeness of the linear relationships between the measured variables was determined using Pearson′s correlation coefficient (*r*).

## 3. Results

The results for the carcass characteristics of the experimental animals have been described previously [2] and summarised in Table 1

The choice of the six muscles studied was intended to obtain a set of muscles representing a variation in tenderness and other quality parameters due to their different anatomical positions, functions and commercial value. Means and standard errors of breed and sex on pH_u_, T_u_, muscle WHC, % purge, WBSF, MFL, IMF, collagen characteristics, and meat colour (*L**, *a**, *b**, chroma and hue-angle) for each of these six muscles are presented in Table 2, Table 3, Table 4, Table 5, Table 6 and Table 7, respectively.

IVG presented higher pH_u_ values (*p* ≤ 0.05) compared to those of BG for LTL, BF, and ST muscles, with SM and SS having a tendency (*p* ≤ 0.10) to show breed differences. Sex differences for pH_u_ were more prominent (*p* ≤ 0.05) for SM, SS, with ST showing both breed and sex differences, and therefore a tendency (*p* ≤ 0.10) to have breed × sex interactions. The IS muscle (~6.1) showed, on average, the highest pH_u_, but no differences between breed and sex. In the muscles where pH_u_ differences were found, the IVG seemed to have the higher pH_u_ compared to BG. When sex differences arose, the wethers always tended to have higher pH_u_ than the bucks. On average, the SS had a pH_u_ of ~5.9, followed by BF and ST with pH_u_ between 5.7 and 5.9.

Although there are some tendencies towards breed and sex differences at 1 d post mortem for some muscles, it is only after 4 d post mortem that significant differences were observed in pressed-out water (WHC). WHC varied between 0.35 to 0.40, measured at 4 d post mortem, but LTL measured 0.43 to 0.45, respectively for BG and IVG wethers, compared to 0.38 and 0.39, respectively for BG and IVG bucks. Significant breed and sex effects for WHC at 4 d post mortem were recorded for SM and SS muscles, although the ratio was not as high as for the LTL. Only IS presented a breed difference for % purge, with that of IVG (0.62–0.82%) significantly lower than that of BG (0.97–1.20%). It was observed that, overall, IS and BF seemed to have lower % purge than that of the other muscles (>1.5%) (Results not shown).

Tenderness-related sex differences were recorded for the BF (MFL 1 and 4 d post mortem) and ST (WBSF 1 d post mortem) muscles, while a tendency (*p* ≤ 0.1) for an interaction between sex and breed was recorded for MFL at 1 d post mortem for the SM muscles (Table 3). The BF wether muscle measured shorter MFL than that of the buck muscle (Table 4). Differences were found between the different muscles (results not shown). Some numerical tenderisation from 1 to 4 d post mortem can be observed in each of the Table 2, Table 3, Table 4, Table 5, Table 6 and Table 7, with SM, SS and IS being the most tender after 4 d post mortem.

All the muscles showed differences by sex (*p* ≤ 0.05) for IMF (Table 2, Table 3, Table 4, Table 5, Table 6 and Table 7). Wether muscles overall recorded a higher percentage IMF than that of bucks in LTL, SM, BF, SS, IS and ST. IVG bucks recorded the lowest values (1.1%) in the IS muscles (Table 6) and BG wethers recorded the highest values of 4.18% in the BF muscle (Table 4). In most muscles, the bucks had about 1% less IMF than that of the wethers, whilst buck BF muscle had up to 2% less IMF than that of its equivalent wether muscle.

There were no significant differences in breed and sex in any of the collagen characteristics among the six muscles studied (Table 2, Table 3, Table 4, Table 5, Table 6 and Table 7). However, there were tendencies (*p* ≤ 0.1) observed for IVG buck LTL as well as BG and IVG buck ST to have higher collagen solubility levels.

Meat colour differences related to sex were noted; *L** (lightness) differences were observed in LTL (1 d post mortem), SM (1 and 4 d post mortem), BF (1 d post mortem), SS (1 and 4 d post mortem), and IS (1 d post mortem), with a trend in the ST for a breed × sex interaction. For these muscles, wethers recorded lower *L** values (darker meat) than the bucks. Sex differences for *a** and chroma (saturation index) were recorded in LTL (1 d post mortem), SM (1 d post mortem), BF (1-d post mortem), and SS (1 d post mortem). These muscles from wethers seem darker and a brighter red than those of bucks, especially at 1 d post mortem. At 4 d post mortem, the hue-angles (discolouration) of wether LTL, SM, SS and IS were lower than that of the corresponding buck muscles. Significant breed * sex interactions were observed for the chroma of the SM and ST, along with a trend in BF at 4 d post mortem indicating towards a higher saturation index for BG wethers and IVG bucks. No breed or sex differences were detected for *b** for any of the muscles.

## 4. Discussion

Compared to extensive studies on the influence of muscle source on meat quality indicators such as pH_u_, chemical composition, tenderness, juiciness and colour attributes in other livestock, only limited studies examined these phenomena in chevon (goat meat); with the focus mainly being on the LTL and SM muscles [8,37,38]. The present study investigated the meat quality of six different muscles: LTL, SM, BF, SS, IS, and ST. It is expected that different muscles will show different results for the various meat quality characteristics [39]. However, when considering differences between breeds and sexes in muscles, few and often negligible differences were found.

Except for the LTL muscle (which is usually the standard position for monitoring pH), pH_u_ for all muscles were in general above 5.8 or even 6.0, suggesting an effect of long-term stress, irrespective of breed or sex. This effect was displayed despite relatively high energy supplementation of the animals during growth and limited pre-slaughter stress due to the short transport distance from the grower facility to abattoir, combined with a short lairage period. IVG animals recorded higher pH_u_ for BF and ST muscles, while the SM, SS and ST muscles of wethers, often of the IVG, were higher than those of bucks, suggesting a slightly higher susceptibility to stress by this breed type when muscles other than the LTL were considered. Similar final pH_u_ values for LTL were reported when high-energy supplements were fed to animals [8,40]. Pophiwa [8] found no differences in LTL pH_u_ between breeds. When animals were kept on natural pasture, it seems that higher final pH values (>5.8) for LTL accompanied by a dark, firm, and dry final meat condition (DFD) prevailed [41,42,43]. A possible reason could be that goats reared on pastures (extensively) have limited, if any, interaction with humans compared to goats reared in feedlot systems, resulting in the former being more stressed because of exposure to humans during the transport, lairage and slaughter processes.

Goats tend to deposit most of their fat in the visceral, rather than carcass depots and produce leaner carcasses [44,45,46] whilst the “indigenous” goat groups usually give inferior results compared to that of the BG [8,9,10]. However, improved feeding conditions in our study benefited castrated animals of both breeds concerning intramuscular fat deposition in all muscles. A faster rate of deposition for carcass and non-carcass fat, as well as total fat, has been reported for does and wethers raised under intensive management compared to bucks [42,43]. Higher marbling levels could contribute to eating quality, although it did not seem to have any effect on WBSF in our or other studies. Shahrai [47] could find no effect of % IMF and WBSF for beef with IMF values between 6.8% and 20.9%. Only when % IMF levels reached 33.9% did the effect become significant [48]. The IMF levels in our study were much lower, between 1.1% and 7.7%. According to Corbin [49], marbling levels varying between 1.96% and 3.8% had no effect on consumers’ scores for beef tenderness (not WBSF), but scores increased significantly at 5.6% and higher.

Variation in connective tissue characteristics across muscles were consistent with the muscle type, which agrees with the findings for cattle [29]. The LTL recorded low total collagen levels while the other muscles recorded much higher levels. Collagen solubility varied between 27% and 39%. MFL, measuring the amount of ageing, varied between 27 and 47 µm [50]. However, the differences in WBSF within muscle were only recorded for 1 d post mortem ST muscle in favour of BG wethers, but were not complimented by any expected differences in MFL or connective tissue characteristics. MFLs of BF muscle were shorter in wethers than rams and only showed numerically lower WBSF values for BG wethers. Under the conditions of this trial, considering the use of electrical stimulation and normal commercial chilling conditions, it can therefore be concluded that neither goat breed nor sex had any effect on WBSF, irrespective of muscle. It is, however, important to note that on average, the LTL muscles had the most advantageous post mortem proteolytic activity (as indicated by the MFL) and lowest total collagen [51], and even though ES was applied during slaughter, the LTL was still tough, as indicated by the high WBSF. No plausible explanation for the tougher LTL muscles could be found, since the choice of a dry cooking method (broiling) also corresponded with the low connective tissue of the cut.

The differences in muscle physiology between species could also explain some of the colour differences noted. Neethling [52] reported muscle-specificity in fresh meat from a medium-sized wild ungulate, the blesbok (*Damaliscus pygargus phillipsi*) and observed that the blesbok *Infraspinatus* muscle was more colour-stable than the LTL and BF. This observation is different from that previously reported for fresh beef [51], and suggests that game species have unique biology and that the influence of muscle source on colour stability is species-dependent [51]. These observations may support the idea that the goat is a unique species, and that chevon should be approached differently from the other better-known red meats such as beef and mutton. In general, the rate and extent of post mortem glycolysis and ultimate pH of the muscle are critical factors that determine goat meat quality, in particular WHC and meat colour [53]. Contrary to Simela [54], no breed differences in meat colour characteristics for the various BG and IVG muscles were observed in this study. For the SM and SS muscles, wethers recorded higher pH_u_ values, which also coincided with slightly darker muscles, i.e., lower *L** values and higher values for *a** and consequently chroma. Incidentally, there were no differences in purge between breed nor sex. In general, this might have been due to the high pH_u,_ as all values were above 5.8, suggesting higher stress susceptibility in these specific animals [55].

Meat from intact male animals (bulls and rams) is generally darker compared to that of females and castrated males [56]. This is in contrast to the present study, where the wethers had darker meat (*L** < 35.0) compared to bucks (*L** > 36.9). Small and sometimes significant differences were found for other colour parameters, where muscles of wethers in most cases tend to show more vivid colours (higher chroma) and lower discolouration (lower hue angle values). It is known that energy status immediately after slaughter has an influence on meat colour (lightness) and tenderness [57,58].

## 5. Conclusions

Knowledge about meat quality of specific indigenous eco-types is limited as studies usually compare nonspecific “indigenous” goats with BG (well described). This study alleviates some misconceptions that exist about the potential quality of “indigenous” goat meat. More muscle meat quality differences were found between bucks and wethers than between BGs and large-frame IVGs consisting of a mixture of the different goat eco-types. This study showed that the muscles of IVG large-frame goats differed minimally from the same muscles derived from BG when finished off in the same feedlot. This study further showed that goat muscles have different characteristics from those of other red meat animals; further research is warranted to better understand these species′ meat quality characteristics and the factors that influence it. More studies should also focus on understanding how to adapt/manage pre- and post-slaughter procedures to produce the best goat meat (chevon) eating experience.

## Figures and Tables

**Figure 1 animals-12-00382-f001:**
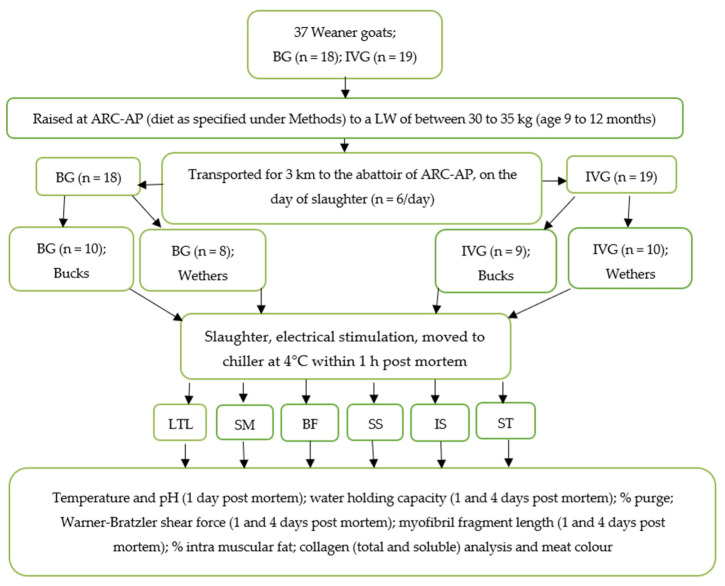
Experimental design to evaluate the effect of the breeds large-frame Indigenous Veld Goat (IVG, Cape Speckled and Cape Lob Ear) and Boer Goat (BG) of southern Africa, on tenderness factors, colour attributes and connective tissue characteristics of *Longissimus thoracis et lumborum* (LTL), *Semimembranosus* (SM), *Biceps femoris* (BF), *Supraspinatus* (SS), *Infraspinatus* (IS), and *Semitendinosus* (ST). ARC-AP = Agricultural Research Council-Animal Production, Irene, South Africa.

**Figure 2 animals-12-00382-f002:**
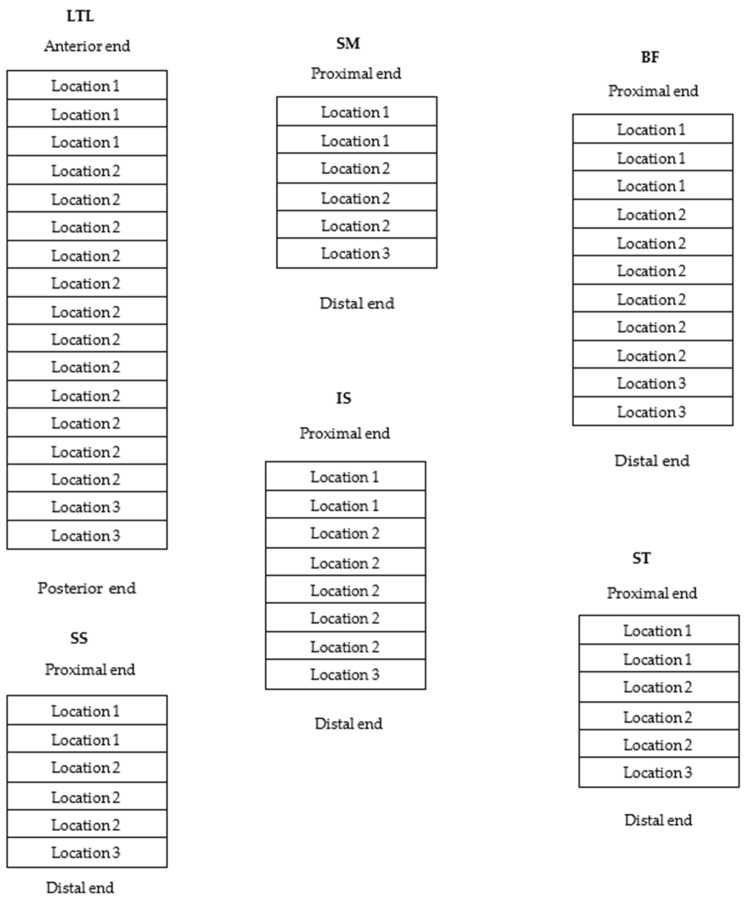
Sampling locations of the six different muscles (i.e., *Longissimus thoracis et lumborum* (LTL), *Semimembranosus* (SM), *Biceps femoris* (BF), *Supraspinatus* (SS), *Infraspinatus* (IS), and *Semitendinosus* (ST)). Left side of carcass for day 1 samples for location 1 (meat colour, water-holding capacity, myofibril fragment length), location 2 (Warner-Bratzler shear force) and location 3 (collagen analysis total and soluble); Right side of carcass for day 4 samples for location 1 (meat colour, water holding capacity, myofibril fragment length), location 2 (Warner-Bratzler shear force) and location 3 (collagen analysis-total and soluble, proximate analysis). Proximal = nearest to the vertebral column. Each horizontal section represents a 2.0 cm-thick slice.

**Table 1 animals-12-00382-t001:** Least square means and standard error (SE) of means for carcass characteristics of Boer- (BG) and large frame Indigenous Veld (IVG) buck and wether goats (adapted from [2]).

Carcass Characteristics	Breed				Significance (*p*-Values)		
	BG		IVG				
	Bucks *n* = 10	Wethers *n* = 8	Bucks *n* = 9	Wethers *n* = 10	Breed	Sex	Breed × Sex
Live weight (kg)	35.40 ^a,b^ ± 4.01	36.13 ^a^ ± 3.02	36 67 ^a^ ± 2.68	32.8 ^b^ ± 2.39	0.293	0.118	0.032
Cold carcass weight (kg)	15.26 ± 2.31	16.25 ± 1.66	15.88 ± 1.83	14.86 ± 0.97	0.541	0.938	0.094
Dressing (%)	42.99 ^a^ ± 2.44	44.95 ^b^ ± 1.08	43.28 ^a^ ± 3.23	45.42 ^b^ ± 2.49	0.508	0.017	0.912

^a,b^ Means in the same row per main effect bearing different letters differ (*p* ≤ 0.05).

**Table 2 animals-12-00382-t002:** Least square means and standard error of means for meat tenderness, meat colour and related physiological characteristics of buck and wether Boer Goats (BG) and Indigenous Veld Goats (IVG) of the *Longissimus thoracis et lumborum* (LTL) muscle.

Meat Quality Characteristics	Breed				Significance (*p*-Values)		
	BG		IVG				
	Bucks	Wethers	Bucks	Wethers	Breed	Sex	Breed × Sex
pHu	5.54 ^a^ ± 0.18	5.60 ^a^ ± 0.05	5.67 ^b^ ± 0.11	5.72 ^b^ ± 0.18	0.011	0.241	0.944
Water holding capacity							
1-dpm ^1^	0.41 ± 0.03	0.39 ± 0.06	0.38 ± 0.04	0.37 ± 0.05	0.101	0.384	0.642
4-dpm	0.38 ^a^ ± 0.04	0.45 ^b^ ± 0.08	0.39 ^a^ ± 0.08	0.43 ^b^ ± 0.07	0.979	0.018	0.515
Purge (%)	1.71 ± 0.84	1.86 ± 0.78	2.00 ± 1.02	1.96 ± 0.79	0.495	0.836	0.721
Warner Bratzler Shear force							
1-dpm (N ^2^)	58.5 ± 1.10	59.0 ± 1.17	57.4 ± 1.15	59.5 ± 1.05	0.958	0.752	0.834
4-dpm (N)	46.5 ± 1.14	40.5 ± 1.12	43.3 ± 0.88	42.9 ± 1.22	0.842	0.395	0.499
Myofibril fragment length							
1-dpm (µm)	37.16 ± 5.46	35.55 ± 4.83	35.26 ± 5.05	37.42 ± 5.04	0.351	0.220	0.319
4-dpm (µm)	33.62 ± 6.21	29.63 ± 2.01	30.32 ± 5.07	29.85 ± 6.14	0.471	0.332	0.426
Marbling ^3^							
IMF (% )	1.97 ^a^ ± 1.11	2.58 ^b^ ± 1.35	1.49 ^a^ ± 0.94	2.59 ^b^ ± 0.70	0.620	0.017	0.473
Collagen characteristics							
Collagen solubility (%)	36.68 ± 10.69	37.55 ± 11.25	38.63 ± 9.83	35.49 ± 11.13	0.973	0.722	0.707
Soluble collagen (mg/g ^4^)	1.37 ^x^ ± 0.58	1.40 ^x^ ± 0.42	1.66 ^y^ ± 0.48	1.27 ^x^ ± 0.38	0.958	0.501	0.080
Insoluble collagen (mg/g)	2.40 ± 0.54	2.50 ± 0.91	2.71 ± 0.42	2.40 ± 0.71	0.549	0.232	0.229
Total collagen (mg/g)	3.68 ± 0.85	3.80 ± 0.85	4.24 ± 0.39	3.59 ± 0.78	0.566	0.222	0.160
Meat colour characteristics							
*L** 1-dpm	35.61 ^a^ ± 2.12	33.50 ^b^ ± 1.20	35.11 ^a^ ± 2.60	33.20 ^b^ ± 2.47	0.877	0.010	0.545
*L** 4-dpm	36.65 ± 3.18	34.75 ± 2.67	35.28 ± 1.35	34.84 ± 2.79	0.755	0.471	0.238
*a** 1-dpm	9.45 ^a^ ± 0.84	11.25 ^b^ ± 0.76	9.90 ^a^ ± 1.60	10.53 ^b^ ± 1.27	0.966	0.004	0.139
*a** 4-dpm	9.75 ± 1.25	10.91 ± 1.12	10.09 ± 0.96	10.43 ± 1.44	0.736	0.168	0.208
*b** 1-dpm	11.16 ± 1.41	11.26 ± 1.18	11.10 ± 1.81	12.14 ± 1.41	0.371	0.236	0.354
*b** 4-dpm	13.04 ± 0.94	12.64 ± 0.65	12.52 ± 0.85	12.48 ± 0.91	0.209	0.413	0.499
Chroma 1-dpm	14.66 ^a^ ± 1.30	15.95 ^b^ ± 1.02	14.93 ^a^ ± 1.96	16.13 ^b^ ± 1.39	0.486	0.015	0.898
Chroma 4-dpm	16.34 ± 1.13	16.74 ± 0.06	16.11 ± 1.10	16.18 ± 1.27	0.340	0.577	0.680
Hue angle 1-dpm	49.58 ^x^ ± 4.02	44.96 ^y^ ± 3.51	48.76 ^x^ ± 6.09	47.74 ^y^ ± 2.73	0.388	0.059	0.139
Hue angle 4-dpm	53.36 ^a^ ± 3.86	49.36 ^b^ ± 2.62	51.16 ^a^ ± 2.39	50.16 ^b^ ± 3.49	0.724	0.026	0.116

^a,b^ Means in the same row per main effect bearing different letters differ (*p* ≤ 0.05). ^x,y^ Means in the same row per main effect bearing different letters was considered a tendency to differ (*p* ≤ 0.1). ^1^ dpm = day post mortem; ^2^ N = Newton; ^3^ Marbling = chemically determined intramuscular fat (IMF); ^4^ mg/g = milligram per gram fresh sample; *L** = lightness; *a** = redness; *b** = yellowness; Chroma = saturation index; Hue angle = discolouration.

**Table 3 animals-12-00382-t003:** Least square means and standard error of means for meat tenderness, meat colour and related physiological characteristics of buck and wether Boer Goat (BG) and Indigenous Veld Goats (IVG) of *Semimembranosus* (SM) muscle.

Meat Quality Characteristics	Breed				Significance (*p*-Values)		
	BG		IVG				
	Bucks	Wethers	Bucks	Wethers	Breed	Sex	Breed × Sex
pH_u_	5.89 ^a^ ± 0.27	5.98 ^a,b^ ± 0.11	5.91 ^a^ ± 0.12	6.17 ^b^ ± 0.25	0.092	0.017	0.267
Water holding capacity							
1-dpm ^1^	0.35 ^x^ ± 0.03	0.35 ^x^ ± 0.03	0.35 ^x^ ± 0.06	0.31 ^y^ ± 0.04	0.205	0.078	0.165
4-dpm	0.35 ^a,b^ ± 0.03	0.35 ^a,b^ ± 0.04	0.36 ^a^ ± 0.06	0.41 ^b^ ± 0.03	0.019	0.026	0.185
Purge (%)	1.89 ± 0.48	2.21 ± 1.12	1.60 ± 1.03	1.92 ± 1.00	0.384	0.306	0.999
Warner Bratzler Shear force							
1-dpm (N ^2^)	37.6 ± 0.44	37.4 ± 0.60	39.7 ± 0.50	35.8 ± 0.71	0.908	0.415	0.230
4-dpm (N)	33.1 ± 0.43	31.9 ± 0.84	34.7 ± 0.49	30.0 ± 0.69	0.968	0.177	0.420
Myofibril fragment length							
1-dpm (µm)	41.06 ± 5.85	45.03 ± 5.03	44.08 ± 4.74	42.13 ± 2.73	0.883	0.560	0.066
4-dpm (µm)	38.64 ± 6.78	37.85 ± 5.78	40.22 ± 3.62	35.46 ± 4.60	0.803	0.130	0.276
Marbling ^3^							
IMF (%)	1.94 ^a^ ± 1.09	3.05 ^b^ ± 1.53	1.76 ^a^ ± 1.05	2.76 ^b^ ± 0.80	0.689	0.008	0.888
Collagen characteristics							
Collagen solubility (%)	35.19 ± 11.59	27.58 ± 9.62	32.91 ± 5.68	33.03 ± 12.27	0.935	0.236	0.572
Soluble collagen (mg/g ^4^)	2.55 ± 1.30	1.76 ± 0.76	2.09 ± 0.53	2.04 ± 1.01	0.602	0.624	0.388
Insoluble collagen (mg/g)	4.43 ± 0.45	4.60 ± 0.67	4.39 ± 0.56	4.11 ± 0.78	0.647	0.207	0.384
Total collagen (mg/g)	6.82 ± 1.60	6.21 ± 1.03	6.32 ± 0.81	5.99 ± 0.97	0.705	0.175	0.467
Meat colour characteristics							
*L** 1-dpm	35.74 ^a^ ± 3.03	33.78 ^b^ ± 1.84	37.24 ^a^ ± 2.36	33.01 ^b^ ± 1.47	0.894	0.0003	0.199
*L** 4-dpm	36.94 ^a^ ± 3.22	34.06 ^b^ ± 2.99	36.33 ^a^ ± 2.08	34.14 ^b^ ± 2.72	0.501	0.012	0.270
*a** 1-dpm	10.55 ^a^ ± 1.40	12.36 ^b^ ± 1.66	10.30 ^a^ ± 1.32	11.74 ^b^ ± 1.72	0.388	0.003	0.060
*a** 4-dpm	9.85 ^a^± 2.03	12.30 ^b^ ± 1.84	11.17 ^b^ ± 1.63	10.37 ^a^ ± 2.21	0.066	0.111	0.018
*b** 1-dpm	11.91 ± 1.31	12.06 ± 1.37	12.31 ± 0.67	12.07 ± 1.31	0.318	0.474	0.580
*b** 4-dpm	12.71 ± 1.21	12.68 ± 0.63	13.26 ± 0.67	12.23 ± 1.38	0.828	0.353	0.512
Chroma 1-dpm	15.99 ^a^ ± 1.49	17.33 ^b^ ± 1.91	16.12 ^a^ ± 0.90	16.89 ^b^ ± 1.84	0.754	0.018	0.375
Chroma 4-dpm	16.14 ^a^ ± 2.06	17.71 ^b^ ± 1.61	17.41 ^b^ ± 1.43	16.16 ^a^ ± 1.99	0.078	0.185	0.024
Hue angle 1-dpm	48.71 ^a^ ± 4.36	44.49 ^b^ ± 3.34	50.39 ^a^ ± 4.10	44.9 ^b^ ± 2.28	0.395	0.001	0.011
Hue angle 4-dpm	52.71 ^a^ ± 4.11	46.21 ± 3.61 ^b^	50.34 ^a^ ± 3.46	48.29 ^b^ ± 4.23	0.215	0.003	0.236

^a,b^ Means in the same row per main effect bearing different letters differ (*p* ≤ 0.05). ^x,y^ Means in the same row per main effect bearing different letters was considered a tendency to differ (*p* ≤ 0.1). ^1^ dpm = day post mortem; ^2^ N = Newton; ^3^ Marbling = chemically determined intramuscular fat (IMF); ^4^ mg/g = milligram per gram fresh sample; *L** = lightness; *a** = redness; *b** = yellowness; Chroma = saturation index; Hue angle = discolouration.

**Table 4 animals-12-00382-t004:** Least square means and standard error of means for meat tenderness, meat colour and related physiological characteristics of buck and wether Boer Goats (BG) and Indigenous Veld Goats (IVG) of *Biceps femoris* (BF) muscle.

Meat Quality Characteristics	Breed				Significance (*p*-Values)		
	BG		IVG				
	Bucks	Wethers	Bucks	Wethers	Breed	Sex	Breed × Sex
pH_u_	5.74 ^a^ ± 0.11	5.71 ^a^ ± 0.14	5.82 ^b^ ± 0.13	5.91 ^b^ ± 0.16	0.003	0.477	0.204
Water holding capacity							
1-dpm ^1^	0.38 ^y^ ± 0.04	0.38 ^y^ ± 0.05	0.36 ^x^ ± 0.04	0.35 ^x^ ± 0.05	0.096	0.550	0.686
4-dpm	0.35 ± 0.04	0.41 ± 0.06	0.37 ± 0.04	0.37 ± 0.06	0.647	0.167	0.074
Purge (%)	0.96 ± 0.34	1.00 ± 0.40	0.97 ± 0.27	0.70 ± 0.35	0.182	0.282	0.188
Warner Bratzler Shear force							
1-dpm (N ^2^)	55.8 ± 1.06	47.1 ± 1.52	49.9 ± 1.09	47.6 ± 1.43	0.444	0.211	0.455
4-dpm (N)	44.5 ± 0.82	34.4 ± 0.78	40.9 ± 0.96	42.1 ± 1.36	0.652	0.213	0.102
Myofibril fragment length							
1-dpm (µm)	43.57 ^a^ ± 9.93	35.01 ^b^ ± 5.51	40.81 ^a^ ± 6.80	38.89 ^b^ ± 6.50	0.989	0.046	0.188
4-dpm (µm)	35.11 ^a^ ± 5.76	28.26 ^b^ ± 3.54	33.29 ^a^ ± 7.04	32.21 ^b^ ± 5.27	0.724	0.044	0.128
Marbling ^3^							
IMF (% Fat)	2.75 ^a^ ± 1.85	4.18 ^b^ ± 2.46	1.88 ^a^ ± 1.29	3.74 ^b^ ± 0.74	0.345	0.005	0.694
Collagen characteristics							
Collagen solubility (%)	37.88 ± 14.34	34.50 ± 7.73	27.93 ± 9.14	37.33 ± 16.13	0.450	0.418	0.143
Soluble collagen (mg/g ^4^)	2.80 ± 1.67	2.46 ± 1.44	1.82 ± 0.78	2.43 ± 1.21	0.218	0.286	0.646
Insoluble collagen (mg/g)	4.27 ± 0.97	4.49 ± 0.87	4.67 ± 0.43	4.09 ± 1.11	0.519	0.505	0.974
Total collagen (mg/g)	6.92 ± 2.22	6.81 ± 2.19	6.33 ± 0.91	6.36 ± 1.25	0.466	0.467	0.938
Meat colour characteristics							
*L** 1-dpm	37.60 ^a^ ± 3.05	33.29 ^b^ ± 2.18	37.11 ^a^ ± 2.38	34.06 ^b^ ± 1.50	0.744	<0.0001	0.246
*L** 4-dpm	38.00 ± 2.56	35.83 ± 1.76	36.68 ± 2.09	36.24 ± 2.93	0.965	0.432	0.160
*a** 1-dpm	9.95 ^a^ ± 1.16	12.29 ^b^ ± 0.99	10.33 ^a,b^ ± 1.62	10.64 ^a,b^ ± 1.41	0.267	0.006	0.027
*a** 4-dpm	8.76 ^a^ ± 1.17	10.84 ^b^ ± 1.36	9.78 ^a,b^ ± 1.33	9.25 ^a,b^ ± 1.19	0.648	0.085	0.004
*b** 1-dpm	11.81 ± 1.33	11.98 ± 1.10	11.89 ± 1.14	12.02 ± 1.52	0.860	0.729	0.997
*b** 4-dpm	11.71 ± 1.31	12.19 ± 1.15	11.84 ± 1.10	11.99 ± 1.23	0.985	0.445	0.671
Chroma 1-dpm	15.49 ^x^ ± 1.40	17.16 ^y^ ± 1.24	15.79 ^x^ ± 1.57	16.11 ^y^ ± 1.67	0.574	0.056	0.179
Chroma 4-dpm	14.66 ^x^ ± 1.62	16.39 ^z^ ± 1.59	15.38 ^y^ ± 1.64	15.23 ^y^ ± 1.29	0.809	0.143	0.072
Hue angle 1-dpm	49.84 ^a^ ± 3.94	44.25 ^b^ ± 2.98	49.21 ^a^ ± 4.49	47.54 ^b^ ± 2.43	0.243	0.005	0.064
Hue angle 4-dpm	53.22 ^b^ ± 2.66	48.95 ^a^ ± 3.26	50.56 ^a,b^ ± 2.33	51.84 ^a,b^ ± 3.73	0.723	0.398	0.010

^a,b^ Means in the same row per main effect bearing different letters differ (*p* ≤ 0.05). ^x,y^ Means in the same row per main effect bearing different letters was considered a tendency to differ (*p* ≤ 0.1). ^1^ dpm = day post mortem; ^2^ N = Newton; ^3^ Marbling = chemically determined intramuscular fat (IMF); ^4^ mg/g = milligram per gram fresh sample; *L** = lightness; *a** = redness; *b** = yellowness; Chroma = saturation index; Hue angle = discolouration.

**Table 5 animals-12-00382-t005:** Least square means and standard error of means for meat tenderness, meat colour and related physiological characteristics of buck and wether Boer Goats (BG) and Indigenous Veld Goats (IVG) of *Supraspinatus* (SS) muscle.

Meat Quality Characteristics	Breed				Significance (*p*-Values)		
	BG		IVG				
	Bucks	Wethers	Bucks	Wethers	Breed	Sex	Breed × Sex
pH_u_	5.89 ^a^ ± 0.27	5.98 ^b^ ± 0.11	5.91 ^a^ ± 0.12	6.17 ^b^± 0.25	0.092	0.017	0.267
Water holding capacity							
1-dpm ^1^	0.35 ^x^ ± 0.03	0.35 ^x^ ± 0.03	0.35 ^x^ ± 0.06	0.31 ^y^ ± 0.04	0.205	0.078	0.165
4-dpm	0.35 ^a,b^ ± 0.03	0.35 ^a,b^ ± 0.04	0.36 ^a^ ± 0.06	0.41 ^b^ ± 0.03	0.019	0.026	0.185
Purge (%)	1.89 ± 0.48	2.21 ± 1.12	1.60 ± 1.03	1.92 ± 1.00	0.384	0.306	0.999
Warner Bratzler Shear force							
1-dpm (N ^2^)	37.6 ± 0.44	37.4 ± 0.60	39.7 ± 0.50	35.8 ± 0.71	0.908	0.415	0.230
4-dpm (N)	33.1 ± 0.43	31.9 ± 0.84	34.7 ± 0.49	30.0 ± 0.69	0.968	0.177	0.420
Myofibril fragment length							
1-dpm (µm)	41.06 ± 5.85	45.03 ± 5.03	44.08 ± 4.74	42.13 ± 2.73	0.883	0.560	0.066
4-dpm (µm)	38.64 ± 6.78	37.85 ± 5.78	40.22 ± 3.62	35.46 ± 4.60	0.803	0.130	0.276
Marbling ^3^							
IMF (%)	1.94 ^a^ ± 1.09	3.05 ^b^ ± 1.53	1.76 ^a^ ± 1.05	2.76 ^b^ ± 0.80	0.689	0.008	0.888
Collagen characteristics							
Collagen solubility (%)	35.19 ± 11.59	27.58 ± 9.62	32.91 ± 5.68	33.03 ± 12.27	0.741	0.297	0.202
Soluble collagen (mg/g ^4^)	2.55 ± 1.30	1.76 ± 0.76	2.09 ± 0.53	2.04 ± 1.01	0.697	0.575	0.179
Insoluble collagen (mg/g)	4.43 ± 0.45	4.60 ± 0.67	4.39 ± 0.56	4.11 ± 0.78	0.498	0.359	0.838
Total collagen (mg/g)	6.82 ± 1.60	6.21 ± 1.03	6.32 ± 0.81	5.99 ± 0.97	0.987	0.946	0.128
Meat colour characteristics							
*L** 1-dpm	35.74 ^a^ ± 3.03	33.78 ^b^ ± 1.84	37.24 ^a^ ± 2.36	33.01 ^b^ ± 1.47	0.649	0.0003	0.222
*L** 4-dpm	36.94 ^a^ ± 3.22	34.06 ^b^ ± 2.99	36.33 ^a^ ± 2.08	34.14 ^b^ ± 2.72	0.991	0.012	0.450
*a** 1-dpm	10.55 ^a^ ± 1.40	12.36 ^b^ ± 1.66	10.30 ^a^ ± 1.32	11.74 ^b^ ± 1.72	0.558	0.003	0.720
*a** 4-dpm	9.85 ^a^ ± 2.03	12.30 ^b^ ± 1.84	11.17 ^a,b^ ± 1.63	10.37 ^a,b^ ± 2.21	0.788	0.224	0.018
*b** 1-dpm	11.91 ± 1.31	12.06 ± 1.37	12.31 ± 0.67	12.07 ± 1.31	0.623	0.885	0.597
*b** 4-dpm	12.71 ± 1.21	12.68 ± 0.63	13.26 ± 0.67	12.23 ± 1.38	0.853	0.131	0.153
Chroma 1-dpm	15.99 ^x^ ± 1.49	17.33 ^y^ ± 1.91	16.12 ^x^ ± 0.90	16.89 ^y^ ± 1.84	0.934	0.054	0.591
Chroma 4-dpm	16.14 ^a^ ± 2.06	17.71 ^a,b^ ± 1.61	17.41 ^a,b^ ± 1.43	16.16 ^b^ ± 1.99	0.911	0.811	0.024
Hue angle 1-dpm	48.71 ^a^ ± 4.36	44.49 ^b^ ± 3.34	50.39 ^a^ ± 4.10	44.9 ^b^ ± 2.28	0.351	0.001	0.934
Hue angle 4-dpm	52.71 ^a^ ± 4.11	46.21 ^b^ ± 3.61	50.34 ^a^ ± 3.46	48.29 ^b^ ± 4.23	0.800	0.003	0.054

^a,b^ Means in the same row per main effect bearing different letters differ (*p* ≤ 0.05). ^x,y^ Means in the same row per main effect bearing different letters was considered a tendency to differ (*p* ≤ 0.1). ^1^ dpm = day post mortem; ^2^ N = Newton; ^3^ Marbling = chemically determined intramuscular fat (IMF); ^4^ mg/g = milligram per gram fresh sample; *L** = lightness; *a** = redness; *b** = yellowness; Chroma = saturation index; Hue angle = discolouration.

**Table 6 animals-12-00382-t006:** Least square means and standard error of means for meat tenderness, meat colour and related physiological characteristics of buck and wether Boer Goats (BG) and Indigenous Veld Goats (IVG) of *Infraspinatus* (IS) muscle.

Meat Quality Characteristics	Breed				Significance (*p*-Values)		
	BG		IVG				
	Bucks	Wethers	Bucks	Wethers	Breed	Sex	Breed × Sex
pH_u_	5.97 ± 0.26	6.11 ± 0.10	6.09 ± 0.24	6.12 ± 0.21	0.324	0.247	0.446
Water holding capacity							
1-dpm ^1^	0.36 ± 0.05	0.38 ± 0.07	0.34 ± 0.05	0.34 ± 0.05	0.195	0.791	0.606
4-dpm	0.35 ± 0.05	0.39 ± 0.06	0.38 ± 0.04	0.37 ± 0.05	0.686	0.419	0.199
Purge (%)	0.97 ^a^ ± 0.35	1.20 ^a^ ± 0.57	0.82 ^b^ ± 0.49	0.62 ^b^ ± 0.23	0.015	0.960	0.129
Warner Bratzler Shear force							
1-dpm (N ^2^)	33.8 ± 0.63	31.9 ± 0.45	29.9 ± 0.40	30.0 ± 0.68	0.155	0.641	0.588
4-dpm (N)	26.9 ^x^ ± 0.37	28.9 ^x^ ± 0.42	25.7 ^y^ ± 0.39	24.8 ^y^ ± 0.54	0.083	0.726	0.331
Myofibril fragment length							
1-dpm (µm)	46.53 ± 6.51	42.70 ± 4.59	44.63 ± 5.51	44.43 ± 8.29	0.886	0.367	0.403
4-dpm (µm)	41..41 ± 7.32	39.36 ± 6.25	38.78 ± 4.06	37.46 ± 5.89	0.232	0.407	0.856
Marbling ^3^							
IMF (%)	1.49 ^a^ ± 0.59	2.70 ^b^ ± 1.10	1.10 ^a^ ± 0.66	2.09 ^b^ ± 0.41	0.092	<0.0001	0.641
Collagen characteristics							
Collagen solubility (%)	37.05 ± 10.26	39.39 ± 9.81	38.31 ± 11.58	34.79 ± 9.69	0.513	0.873	0.538
Soluble collagen (mg/g ^4^)	2.83 ± 1.14	2.76 ± 1.22	2.89 ± 1.11	2.33 ± 0.74	0.793	0.396	0.200
Insoluble collagen (mg/g)	4.89 ± 1.06	4.18 ± 1.15	4.67 ± 0.81	4.47 ± 0.81	0.848	0.133	0.733
Total collagen (mg/g)	7.55 ± 1.55	6.79 ± 2.04	7.39 ± 1.21	6.61 ± 0.89	0.891	0.131	0.598
Meat colour characteristics							
*L** 1-dpm	36.96 ^a^ ± 3.39	34.64 ^b^ ± 2.57	38.36 ^a^ ± 2.32	37.0 ^b^ ± 2.15	0.057	0.048	0.537
*L** 4-dpm	37.61 ± 3.34	36.03 ± 2.64	38.21 ± 2.43	37.19 ± 3.88	0.461	0.221	0.785
*a** 1-dpm	8.22 ^a^ ± 1.92	10.28 ^b^± 1.45	8.43 ^a^ ± 1.64	9.03 ^b^ ± 2.18	0.519	0.040	0.244
*a** 4-dpm	8.86 ^a^ ± 1.70	10.84 ^b^ ± 2.08	8.69 ^a^ ± 1.74	9.60 ^b^ ± 2.42	0.402	0.039	0.447
*b** 1-dpm	10.60 ^a^ ± 1.52	10.89 ^a^ ± 1.47	12.13 ^b^ ± 0.71	11.17 ^b^ ± 0.97	0.042	0.815	0.411
*b** 4-dpm	12.41 ± 1.28	12.20 ± 1.34	12.46 ± 1.22	11.98 ± 1.02	0.831	0.364	0.712
Chroma 1-dpm	13.52 ± 2.18	15.03 ± 1.98	14.53 ± 1.98	14.48 ± 1.84	0.642	0.289	0.254
Chroma 4-dpm	15.36 ± 1.67	16.36 ± 2.20	15.32 ± 1.72	15.54 ± 1.86	0.545	0.332	0.508
Hue angle 1-dpm	52.73 ^a^ ± 4.83	47.01 ^b^ ± 2.23	55.90 ^a^ ± 4.28	50.57 ^b^ ± 4.40	0.034	0.001	0.586
Hue angle 4-dpm	54.86 ± 5.07	49.16 ± 4.10	55.66 ± 4.87	52.18 ± 7.07	0.409	0.017	0.544

^a,b^ Means in the same row per main effect bearing different letters differ (*p* ≤ 0.05). ^x,y^ Means in the same row per main effect bearing different letters was considered a tendency to differ (*p* ≤ 0.1). ^1^ dpm = day post mortem; ^2^ N = Newton; ^3^ Marbling = chemically determined intramuscular fat (IMF); ^4^ mg/g = milligram per gram fresh sample; *L** = lightness; *a** = redness; *b** = yellowness; Chroma = saturation index; Hue angle = discolouration.

**Table 7 animals-12-00382-t007:** Least square means and standard error of means for meat tenderness, meat colour and related physiological characteristics of buck and wether Boer Goats (BG) and Indigenous Veld Goats (IVG) of *Semitendinosus* (ST) muscle.

Meat Quality Characteristics	Breed				Significance (*p*-Values)		
	BG		IVG				
	Bucks	Wethers	Bucks	Wethers	Breed	Sex	Breed × Sex
pH_u_	5.66 ^a^ ± 0.11	5.69 ^a^ ± 0.06	5.71 ^b^ ± 0.13	5.89 ^b^ ± 0.18	0.004	0.021	0.091
Water holding capacity							
1-dpm ^1^	0.37 ± 0.04	0.35 ± 0.05	0.38 ± 0.03	0.37 ± 0.04	0.432	0.394	0.705
4-dpm	0.38 ± 0.07	0.39 ± 0.06	0.39 ± 0.04	0.41 ± 0.05	0.265	0.421	0.750
Purge (%)	1.49 ± 0.97	1.62 ± 0.83	1.93 ± 1.53	1.54 ± 0.92	0.624	0.708	0.479
Warner Bratzler Shear force							
1-dpm (N ^2^)	50.8 ^a^ ± 0.51	44.8 ^b^ ± 0.48	44.8 ^b^ ± 0.48	44.1 ^b^ ± 1.19	0.440	0.047	0.736
4-dpm (N)	47.3 ± 0.61	41.4 ± 0.32	43.0 ± 0.64	40.8 ± 1.23	0.288	0.137	0.483
Myofibril fragment length							
1-dpm (µm)	46.48 ± 4.56	45.63 ± 3.40	44.06 ± 5.03	46.66 ± 5.38	0.662	0.553	0.274
4-dpm (µm)	40.58 ± 5.24	38.44 ± 4.41	40.12 ± 6.19	38.51 ± 8.17	0.864	0.371	0.899
Marbling ^3^							
IMF (%)	2.12 ^a^ ± 1.53	2.76 ^b^ ± 1.50	1.84 ^a^ ± 1.07	2.93 ^b^ ± 0.68	0.980	0.040	0.590
Collagen characteristics							
Collagen solubility (%)	37.09 ± 11.22	33.60 ± 9.82	35.31 ± 7.75	32.94 ± 8.66	0.821	0.404	0.690
Soluble collagen (mg/g ^4^)	1.85 ^x^ ± 0.52	1.41 ^y^ ± 0.47	1.74 ^x^ ± 0.75	1.57 ^y^ ± 0.55	0.058	0.059	0.757
Insoluble collagen (mg/g)	3.36 ± 1.07	2.89 ± 0.52	3.10 ± 0.30	3.21 ± 0.47	0.688	0.128	0.136
Total collagen (mg/g)	5.08 ± 1.05	4.20 ± 0.57	4.72 ± 0.10	4.70 ± 0.71	0.823	0.104	0.160
Meat colour characteristics							
*L** 1-dpm	40.11 ^x^ ± 2.05	38.73 ^y^ ± 1.68	39.36 ^y^ ± 0.98	39.46 ^y^ ± 2.62	0.963	0.882	0.090
*L** 4-dpm	39.89 ± 2.21	39.58 ± 2.99	39.52 ± 1.68	38.28 ± 3.03	0.781	0.849	0.899
*a** 1-dpm	7.58 ^b^ ± 1.22	9.25 ^b^ ± 0.94	8.17 ^b^ ± 0.85	7.63 ^a^ ± 1.27	0.342	0.891	0.005
*a** 4-dpm	7.21 ^a^ ± 1.28	8.50 ^b^ ± 1.63	8.09 ^a^ ± 1.06	8.96 ^b^ ± 1.61	0.347	0.029	0.392
*b** 1-dpm	12.40 ± 0.78	12.79 ± 1.09	12.84 ± 0.89	12.76 ± 0.73	0.428	0.618	0.408
*b** 4-dpm	12.47 ± 0.91	12.73 ± 0.93	12.80 ± 1.23	13.23 ± 0.80	0.178	0.285	0.785
Chroma 1-dpm	14.64 ^a^ ± 0.93	15.89 ^b^ ± 1.11	15.60 ^b^ ± 0.82	14.79 ^a^ ± 1.08	0.959	0.594	0.004
Chroma 4-dpm	14.47 ^x^ ± 1.23	15.41 ^y^ ± 1.33	15.19 ^x^ ± 1.49	15.86 ^y^ ± 0.90	0.110	0.059	0.744
Hue angle 1-dpm	59.04 ^a,b^ ± 4.34	54.45 ^a^ ± 3.42	58.40 ^a,b^ ± 3.14	59.51 ^b^ ± 3.76	0.236	0.936	0.029
Hue angle 4-dpm	60.12 ± 4.16	56.94 ± 5.15	58.16 ± 2.41	55.96 ± 5.36	0.671	0.335	0.421

^a,b^ Means in the same row per main effect bearing different letters differ (*p* ≤ 0.05). ^x,y^ Means in the same row per main effect bearing different letters was considered a tendency to differ (*p* ≤ 0.1). ^1^ dpm = day post mortem; ^2^ N = Newton; ^3^ Marbling = chemically determined intramuscular fat (IMF); ^4^ mg/g = milligram per gram fresh sample; *L** = lightness; *a** = redness; *b** = yellowness; Chroma = saturation index; Hue angle = discolouration.

## Data Availability

Except for the carcass quality, supportive data provided in Table 1 that was published in Van Wyk et al. (2020), none of the project related data was published yet. Van Wyk, G.L. did complete a thesis on the full project from which additional publications will follow. Van Wyk, G.L.; Hoffman, L.C.; Strydom, P.E.; Frylinck, L. Effect of Breed Types and Castration on Carcass Characteristics of Boer and Large Frame Indigenous Veld Goats of Southern Africa. Animals 2020, 10, 1884. https://doi.org/10.3390/ani10101884.

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
