# Peer review of "Differences in Meat Quality of Six Muscles Obtained from Southern African Large-Frame Indigenous Veld Goat and Boer Goat Wethers and Bucks"

_animals, 2022, doi:10.3390/ani12030382_

Round 1

Reviewer 1 Report

It is a well-written study that does a fantastic job of showcasing the two breeds' meat qauality parameters. I have the following suggestions for the author's consideration:

Both breeds had a total sample size of 37 goats, 18 BG, and 19 IVG. Could you perhaps elaborate on the sires of these two breeds? Because this sample represents a representation of the two breeds, we need to be certain about their sires. Do all of these offspring come from one or more of these two breeds' sires?

Figure 2 is excellent, however, could you consider using the goat diagram and then highlighting/marking the position of specific muscles (rather than all boxes)?

Tables 2-7: There is a large quantity of data in this table, making it difficult for the reader to rapidly absorb the information. Can you create a separate table with just significant data for each of the six muscles? Additionally, information from tables 2-7 might be presented as extra information.

The discussion is still quite broad and hazy. Could you perhaps make it more readable and concrete for the readers?

Author Response

Response to Reviewer 1 Comments

Thank you for your positive comments. Highly appreciated.

Reviewer 2 Report

Thank you for the interesting article, but I have a few comments about it:
- in my opinion the title should be modified: Differences in tenderness and colour of six muscles obtained from Southern African Large Frame Indigenous Veld Goat and Boer Goat wethers and bucks
- if the meat of wethers and bucks is compared, should the term "sex" be used?
- were the individual muscles weighed? It could be important for consumers
- in the methodology, it is also worth mentioning that the IMF analysis concerns the marbling of meat
- line 15-16- this sentence is unnecessary here
- line 50- "The major reason of that is ...."
- linijka 62- 63- "... many atributes such as texture ....., color, which are important to consumers"
- line 99 - was the pellet dosed individually?
- minor errors in the bibliography (e.g. lack of bold) in items: 9, 31, 39 and 47

Reviewer 3 Report

This work is interesting because it presents data from a poorly studied local breed of goats in South Africa. However, I am afraid that may limit the interest of the scientific community in general.

The work is well organized in terms of description of the experimental work and presentation of results.

The title is too long and, even so, it does not represent the experimental work carried out. It would be better to replace “tenderness and colour” by “meat quality”. In this experimental work, much more than just tenderness and colour has been determined.

Line 96 could indicate what kind of hay the animals were offered and, if possible, its protein and fiber content (e.g. NDF).

The title of section 2.2.4. it should just be “Chemical composition and characteristics of the collagen”.

The title of point 2.2.5. should be replaced by "measurement of color and pH".

In line 208 the “%DL” is the same as "purge" described in section 2.2.1. The same term should be used throughout the text.

The abbreviation of the names of the 6 muscles studied must be used throughout the text, and must be placed on lines 164 and 165; lines 298, 299 and 300; lines 406, 407 and 408. The same is true for all other abbreviations (e.g. WBST, WHC, MFL, etc.), you don’t have to keep repeating them.

The titles of the various sections of the Material and Methods must not include abbreviations.

The name of the muscles must always appear in italics, therefore the titles in tables 3, 4 and 5 must be corrected.

The conclusions are clear and well suited to the results obtained.

Reference number 8 (line 532) has the wrong link (doi).
